# Strategies for high cell density cultivation of *Akkermansia muciniphila* and its potential metabolism

Haiting Wu,[1] Shuhua Qi,[1] Ruixiong Yang,[1] Qihua Pan,[1] Yinghua Lu,[1,2,3] Chuanyi Yao,[1,2] Ning He,[1,2] Song Huang,[4] Xueping Ling[1,2]

**ABSTRACT** *Akkermansia muciniphila* (*A. muciniphila*) has sparked widespread interest as a potential probiotic bacterium with many physiological functions that colonizes the human intestinal tract. The development of its *in vitro* culture is a promising and urgent research direction. Therefore, culture conditions were first optimized, and a significant improvement of cell density of *A. muciniphila* was achieved in a shake flask with 15.0 g/L of glucose, 37.0 g/L of tryptone, and an initial pH of 7.8. A high $OD_{600}$ (optical density at 600 nm) value of 13.03 ($1.03 \times 10^{10}$ CFU/mL) was reached in a 5-L bioreactor by stage pH controlling, which is the highest reported value by use of a sole carbon source (glucose). Analysis of cell characteristics and protein expression showed that the optimized culture did not affect cellular morphology and the expression of the special outer membrane functional protein (Amuc_1100), while remarkably improving cell hydrophobicity, which is beneficial for bacterial colonization of the gut. The pattern of supernatant metabolites indicated that the optimized medium may promote cell reproduction by inducing cells to produce dethiobiotin and strengthen the metabolic pathway of glycerol 3-phosphate by shifting more glyceraldehyde 3-phosphate dehydrogenase toward secretion at the cell surface, thereby improving cell surface hydrophobicity and adhesion to mucin. This study accomplished the high cell density culture of *A. muciniphila* without affecting its biological function, which also provides a more conducive fermentation strategy to enhance cell adhesion and facilitates its colonization in the gut.

**IMPORTANCE** Currently, there is significant interest in *Akkermansia muciniphila* as a promising next-generation probiotic, making it a hot topic in scientific research. However, to achieve efficient industrial production, there is an urgent need to develop an *in vitro* culture method to achieve high biomass using low-cost carbon sources such as glucose. This study aims to explore the high-density fermentation strategy of *A. muciniphila* by optimizing the culture process. This study also employs techniques such as LC-MS and RNA-Seq to explain the possible regulatory mechanism of high-density cell growth and increased cell surface hydrophobicity facilitating cell colonization of the gut *in vitro* culture. Overall, this research sheds light on the potential of *A. muciniphila* as a probiotic and provides valuable insights for future industrial production.

**KEYWORDS** *Akkermansia muciniphila*, high cell density, cell surface hydrophobicity, out-membrane functional protein, metabolites

A kkermansia muciniphila is a potential next-generation probiotic bacterium isolated from healthy human feces using mucin as a source of carbon and nitrogen in the human gut. Studies have shown that *A. muciniphila* has many functions, such as regulating the intestinal microecological balance and human metabolism associated with obesity (1, 2), inflammatory bowel disease (3, 4), diabetes (5), hypertension (6), autism (7), progeria (8), epilepsy (9), Parkinson disease (10), ALS (11), etc., which indicates that it has broad application prospects.

Address correspondence to Xueping Ling, xpling@xmu.edu.cn.

Haiting Wu and Shuhua Qi contributed equally to this article. Author order was determined alphabetically.

The authors declare no conflict of interest.

See the funding table on p. 17.

The *in vitro* culture of *A. muciniphila* is faced with the problem of low substrate utilization and biomass. *A. muciniphila* can degrade the majority (85%) of mucin in the human gut through a variety of extracellular and intracellular glycosidases (12). Given the difficult availability of human-derived mucin and the structural similarity between human-derived mucin and porcine gastric mucin, Derrien et al. (13) first used porcine gastric mucin and sterile rumen fluid instead of human-derived mucin to successfully isolate and culture *A. muciniphila*. A mucin-based medium has been the first choice for culturing *A. muciniphila* since then. Li et al. (14) tested the effects of different prebiotics on the *in vitro* culture of *A. muciniphila* and found that mulberry galacto-oligosaccharide had the greatest effect on promoting growth. Yin et al. (15) reported that adding 0.4 g/L tryptophan could increase cell viability and accelerate the division rate. Considering that the use of mucin in the medium is complicated and costly, it is not conducive to industrial-scale culture. Belzer and de Vos (16) modified the mucin medium further using 32 g/L of soy peptone plus 25 mmol/L of glucose and N-acetyl-glucosamine (GlcNAc) instead of animal-deriveenzymes of GlcNAc precursor d mucin, obtaining an optimal growth ($OD_{600}$ = 2.5). The same authors reported that cell growth could be stimulated by adding a certain amount of threonine or yeast extract (YE). When a mixed culture of glucose and GlcNAc was used in a 1-L fermenter, an $OD_{600}$ of 7.2 could be achieved. Sharon (17) used pea peptone as a nitrogen source (32 g/L), and the biomass reached its highest reported level ($OD_{600}$ = 15) when the ratio of glucose to GlcNAc is 3:1. For the industry, GlcNAc as a fermentation carbon source is relatively uneconomical, and compared to the culture density produced by other probiotics (18, 19), the large-scale, low-cost, and efficient cultivation of *A. muciniphila* is still worth further exploration.

To achieve a high cell density culture of *A. muciniphila*, the present study optimized the culture conditions in a shake flask and carried out the scale-up culture in a 5-L bioreactor using a pH-controlling strategy. Moreover, the effects of the optimized culture on cell characteristics, protein expression, and metabolites were investigated. This study will provide strategies for attaining *A. muciniphila* industrial production.

## RESULTS AND DISCUSSION

### Effects of the carbon/nitrogen source and culture conditions on *A. muciniphila*

#### Cell growth of A. muciniphila under different carbon sources

Ten different carbon sources were selected in this experiment instead of mucin. As shown in Fig. 1A, *A. muciniphila* grew only in some carbon sources, namely, mucin, glucose, GlcNAc, oligogalactose, fucose, and mannose. Among these, the mucin culture reached the highest $OD_{600}$ value of 2.22, followed by GlcNAc and glucose, which could both reach about 50% of the highest $OD_{600}$ value of mucin. GlcNAc is an essential component of mucin (19), which may explain why the cell growth was lower than that of mucin alone. The human intestine still contains high amounts of glucose, stressing *A. muciniphila*, which can still use glucose (20). In addition, glucose is a monosaccharide that is easier to absorb than GlcNAc and can be quickly used for energy production and, thus, promotes cell growth. *A. muciniphila* can also use oligogalactose, fucose, and mannose, but their use is relatively weak. No bacterial growth was observed in the fermentation broth containing xylose, galactose, sucrose, and glycerol as the carbon source. Since glucose is cheaper than GlcNAc and mucin and favorable for future industrial production, it was selected for the next optimization.

As shown in Fig. 1B, the cell density increased with the enhancement of glucose concentration and reached the highest $OD_{600}$ value (3.10) at 10.0 g/L glucose, which increased by 39.64% compared to that of the mucin medium (Fig. 1A, $OD_{600}$ = 2.22). This indicated that moderate glucose can be effectively absorbed by *A. muciniphila* for energy production, promoting cell growth. Higher glucose concentrations (15.0 and 20.0 g/L) exhibited no further promoting effect on cell growth. There was a lot of residual glucose in the medium. Therefore, 10.0 g/L glucose was selected as the best carbon source.

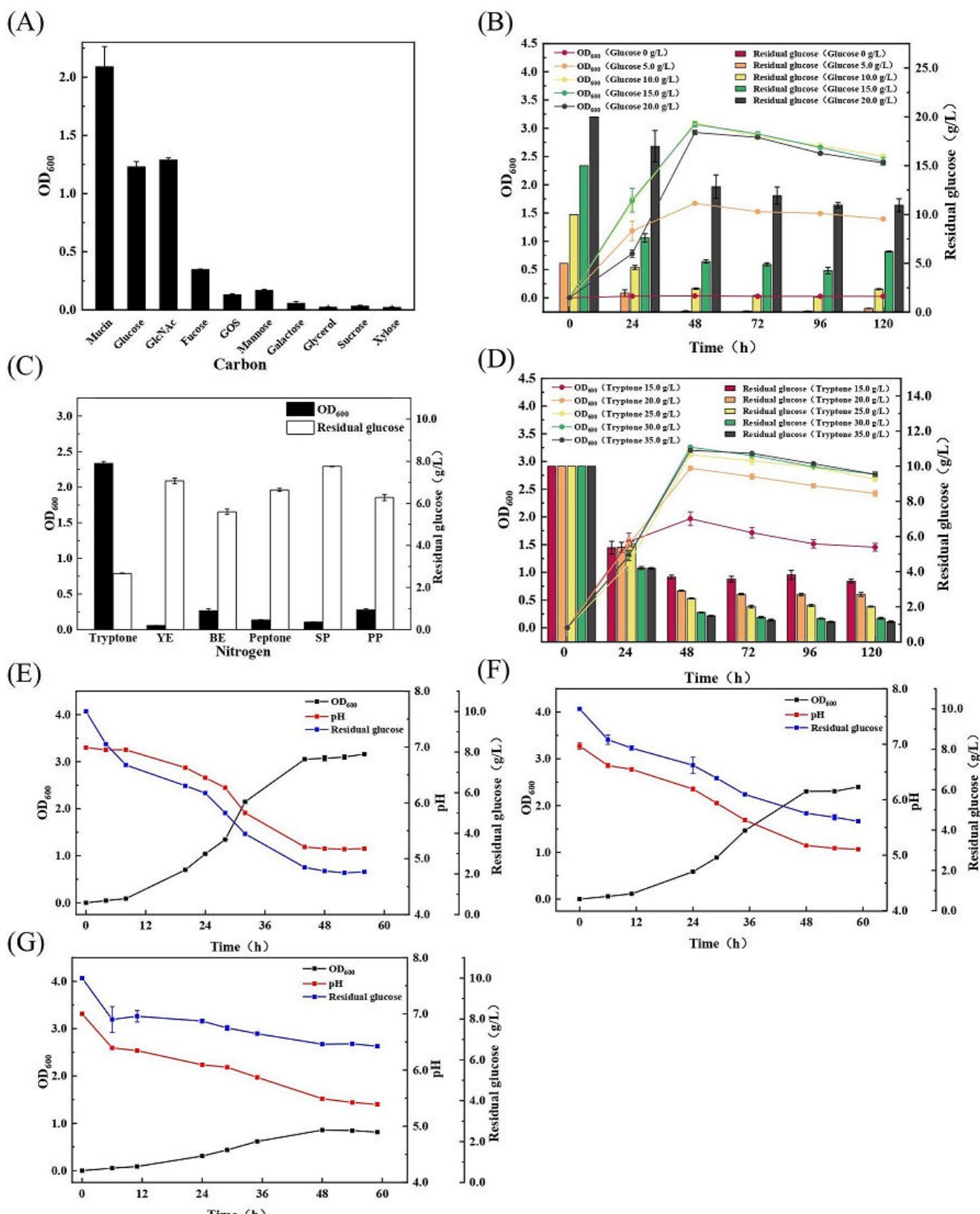

**FIG 1** Cells of *A. muciniphila* grown in different types of carbon sources (A), concentrations of glucose (B), types of nitrogen sources (C), concentrations of tryptone (D), and types of buffer solutions including NaHCO$_3$ (E), 3-(4-morpholino) propanesulphonic acid (MOPS) (F), and KH$_2$PO$_4$/Na$_2$HPO$_4$·12H$_2$O (G).

## Cell growth of *A. muciniphila* under different nitrogen sources

As demonstrated in Fig. 1C. *A. muciniphila* grew significantly better in tryptone than in other nitrogen sources, which had a lower density of peptone, proteose peptone (PP), and beef extract (BE), and had almost no growth in YE and soybean peptone (SP). Tryptone is mostly derived from the hydrolysate of casein, the main protein of human milk. It has been shown that *A. muciniphila* can use human milk well, and a small amount of *A. muciniphila* has also been detected in human milk (21), which may explain why *A. muciniphila* grew better in tryptone than in other nitrogen sources. Hence, tryptone was chosen as the nitrogen source for the medium.

The results of the concentration optimization of tryptone are illustrated in Fig. 1D. The biomass gradually increased with increasing tryptone concentration, and the $OD_{600}$ was 3.12 at 25.0 g/L tryptone. At a tryptone concentration of more than 25.0 g/L, the enhancement of biomass was not obvious, and the residual glucose presented the same concentration. Finally, 25.0 g/L tryptone was selected as the best nitrogen source. It is reported that *A. muciniphila* did not encode the key gene glucosamine synthase for the synthesis of peptidoglycan from glucose, so GlcNAc is needed to support the growth of *A. muciniphila* (22). Why do cells of *A. muciniphila* grow well in the medium with carbon and nitrogen sources of glucose and tryptone? According to the preliminary proteomics analysis (Table S1), the related synthesizing enzymes of GlcNAc precursor in *A. muciniphila* are expressed, which indicates that tryptone may contain potential amino sugar precursors or amino sugars to support the synthesis of peptidoglycan and maintain cell growth under a single carbon source.

## Cell growth of *A. muciniphila* under different buffer systems

*A. muciniphila* uses anaerobic metabolic pathways for capacity production, and its growth process produces a large number of organic acids, leading to a decrease in the pH of the culture medium, which is unfavorable for bacterial growth. To reduce the adverse effects of pH decrease, three buffer systems of $NaHCO_3$, MOPS, and $KH_2PO_4$/$Na_2HPO_4 \cdot 12H_2O$ were compared.

Figure 1E through G shows that cell growth in the $NaHCO_3$ buffer system with an $OD_{600}$ of 3.05 was the best, followed by MOPS with an $OD_{600}$ of 2.30. On the other hand, the $KH_2PO_4$/$Na_2HPO_4 \cdot 12H_2O$ buffer system had only an $OD_{600}$ of 0.86, which is probably because the $KH_2PO_4$/$Na_2HPO_4 \cdot 12H_2O$ buffer system contained too many potassium ions, inhibiting the growth of bacteria. The $NaHCO_3$ and MOPS buffer systems consumed 7.91 and 5.57 g/L glucose, respectively, and the corresponding final pH values were 5.18 and 5.11, respectively. These results show that the glucose consumption in the $NaHCO_3$ buffer system was higher than that in MOPS, but its final pH was slightly higher than that in MOPS, indicating that $NaHCO_3$ could better regulate the medium pH reduction compared to MOPS, allowing *A. muciniphila* to grow well. The $NaHCO_3$ buffer system is similar to the buffer system in the human body, and the release of mucin in the human intestine also depends on the bicarbonate-rich environment (23). These observations suggest that *A. muciniphila* has adapted to the bicarbonate-containing environment in the human intestine. Consequently, the $NaHCO_3$ buffer system is more favorable for the growth of *A. muciniphila*.

## Cell growth of *A. muciniphila* under different initial pHs

The pH has a strong influence on cell growth. As shown in Fig. 2A, *A. muciniphila* could grow in the range of the initial pH of 6.5–8.5, but the biomass decreased rapidly after 28 and 34 h at pH values of 8.0 and 8.5. Cells began to enter the logarithmic growth phase after 10 h (Fig. 2A). When the initial pH was between 6.5 and 7.5, the higher the pH is, the faster the cells grew. An initial pH of 7.5 gave the fastest growth rate, reaching the highest OD (3.25) at 34 h, which is higher than that of the initial pH of 6.5 and 7.0. When the initial pH increased to 8.0 and 8.5, the growth rate showed a similar trend to that of pH 7.5 in the first 28 h and presented the highest values (lower than the initial

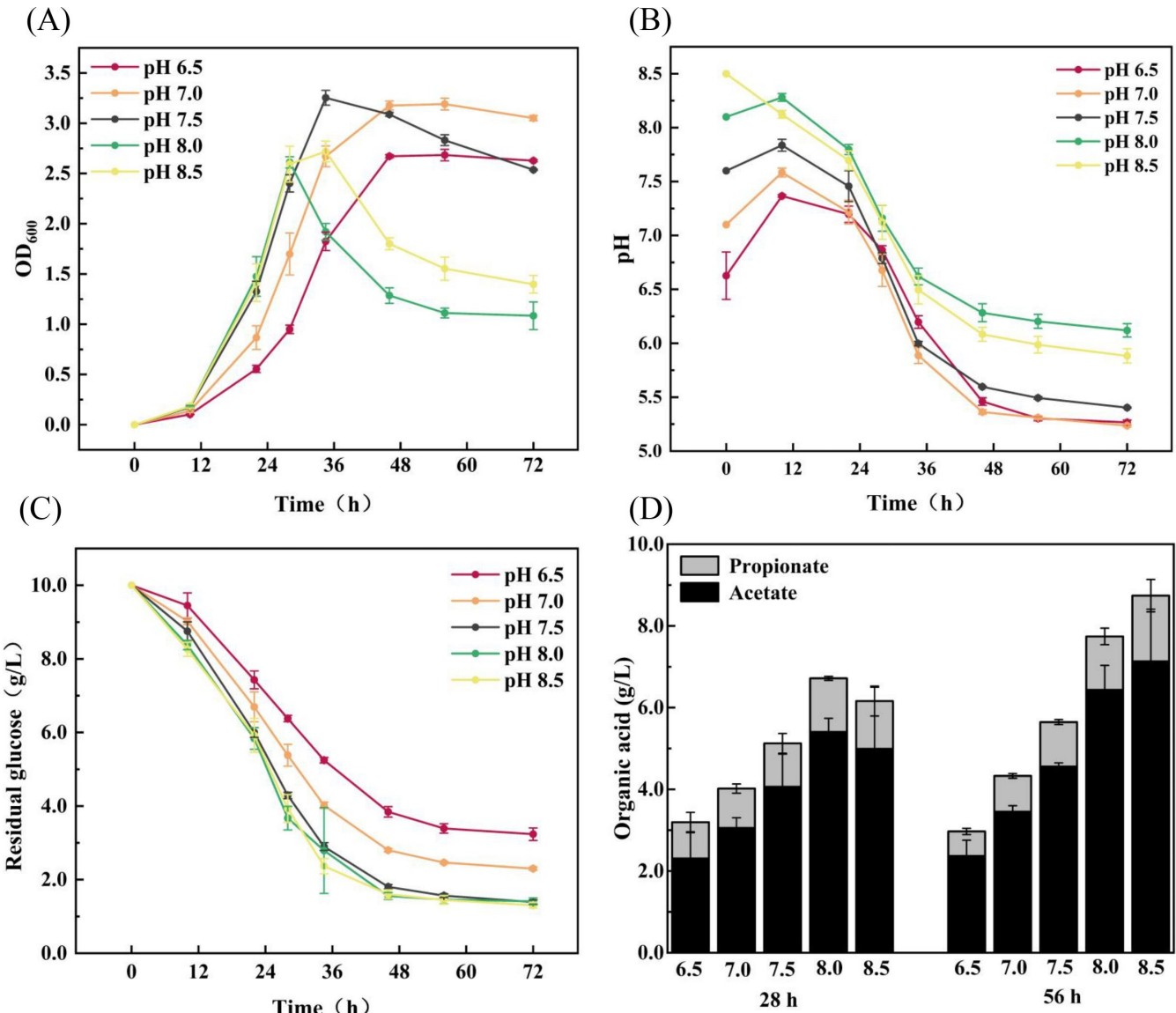

**FIG 2** Cell growth in the medium with different initial pHs (A), pH changes of the medium (B), residual glucose concentration of the medium (C), and organic acid metabolism (D) during cell growth.

pH 7.5) at 28 and 34 h, respectively, and then decreased sharply. It can be seen that the best cell growth was achieved at an initial pH of 7.5. Prior studies have shown that a slightly alkaline environment is more favorable for *A. muciniphila* gut colonization (24, 25). Therefore, a slightly alkaline environment (initial pH 7.5) is appropriate for culturing *A. muciniphila* in a shake flask.

Figure 2A shows that the delay period of *A. muciniphila* ranged from 0 to 12 h. As seen in Fig. 2B, different initial pHs obviously changed the medium pH during the delay phase. When the initial pH was lower than 8.5, the medium pH increased remarkably during the lag phase. The lower the initial pH, the greater the increase in the medium pH. Moreover, the final lag phase pH was concentrated in the range of 7.3–8.2. With an initial pH of 8.5, the medium pH decreased, reaching 8.1 at the final lag phase. Figure 2C also shows that the greater the initial pH, the more glucose is consumed during the lag phase, indicating that the change in the initial pH might affect the pH adaptation of *A. muciniphila* by adjusting the metabolism during the lag phase to make it slightly alkaline for the optimal growth of the organism.

Ottman et al. (26) found that the biomass of *A. muciniphila* declined sharply in the later phase in the medium without mucins, while the addition of aminosaccharides could control the decrease. In the present study, the adjustment of the initial pH of the medium could also control the decline of the biomass of the latter phase (Fig. 2A). Under different initial pHs, cell growth and glucose consumption showed a similar change trend from 0 to 28 h. After 28 h, the biomass of the initial pH of 8.0 and 8.5 began to drop sharply, while it did not exhibit this phenomenon under other pH values. Interestingly, the initial pH values of 8.0 and 8.5 showed the same glucose consumption rate as that of the initial pH of 7.5 after 28 h, which was higher than that of the initial pH values of 6.5 and 7.0 (Fig. 2C). Based on the results in Fig. 2D, the initial pH values of 8.0 and 8.5 accumulated more organic acids at 28 and 56 h than the other pH values. It is speculated that the glucose consumed by the cells after 28 h under the initial pH values of 8.0 and 8.5 is used for the accumulation of organic acids instead of cell growth, suggesting that the over-alkaline initial pH will induce *A. muciniphila* to produce more organic acids. Othman et al. had reported that excessive accumulation of organic acids was detrimental to the growth of lactic acid bacteria (27). Therefore, a large accumulation of organic acids may be adverse to the growth of *A. muciniphila*.

The results mentioned above revealed that adjusting the medium pH might ameliorate the problem of decreasing biomass in the latter stage, which provides the concept for the succeeding bioreactor culture. That is, in the early stage, a slightly alkaline environment is maintained to promote cell growth, whereas in the latter stage, a slightly acidic environment is maintained to prevent excessive organic acid accumulation.

## Box–Behnken design for *A. muciniphila* culture

According to the aforementioned results, three factors that have a greater influence on the growth of *A. muciniphila*, namely, carbon source of glucose, nitrogen source of tryptone, and initial pH, were selected to design a three-factor, three-level response surface test using Design Expert. The results are presented in Table S2.

Quadratic regression analysis was performed based on the above data. The following equation was obtained:

$$Y = 3.07 + 0.57 \times A + 0.38 \times B - 0.27 \times C + 0.31 \times A \times B - 0.44 \times A \times C + 0.277 \times B \times C - 0.67 \times B^2 - 0.29 \times C^2$$

where $Y$ is $OD_{600}$ and $A$, $B$, and $C$ are glucose, tryptone, and initial pH, respectively.

TABLE 1  ANOVA for response surface quadratic model

| Source | Sum of square | df | Mean square | F | Pr > F | |
|---|---|---|---|---|---|---|
| Model | 7.58 | 9 | 0.84 | 36.57 | <0.0001 | Significant |
| *A* | 2.58 | 1 | 2.58 | 112.19 | <0.0001 | |
| *B* | 1.18 | 1 | 1.18 | 51.22 | 0.0002 | |
| *C* | 0.59 | 1 | 0.59 | 25.44 | 0.0015 | |
| *AB* | 0.37 | 1 | 0.37 | 16.25 | 0.0050 | |
| *AC* | 0.0077 | 1 | 0.0077 | 0.33 | 0.5812 | |
| *BC* | 0.29 | 1 | 0.29 | 12.74 | 0.0091 | |
| $A^2$ | 1.91 | 1 | 1.91 | 82.91 | <0.0001 | |
| $B^2$ | 0.12 | 1 | 0.12 | 5.26 | 0.0555 | |
| $C^2$ | 0.34 | 1 | 0.34 | 14.92 | 0.0062 | |
| Residual | 0.16 | 7 | 0.023 | | | |
| *Lack of fit* | 0.10 | 3 | 0.034 | 2.39 | 0.2094 | Not significant |
| *Pure error* | 0.058 | 4 | 0.014 | | | |
| Cor total | 7.74 | 16 | | | | |

**TABLE 2** The verification of quadratic polynomial equation model

| Run | Factors | | | Predicted OD$_{600}$ | Actual OD$_{600}$ |
|---|---|---|---|---|---|
| | *A*: glucose (g/L) | *B*: tryptone (g/L) | *C*: pH | | |
| 1 | 12.24 | 25.24 | 7.26 | 3.28 | 3.12 ± 0.04 |
| 2 | 14.88 | 37.65 | 7.83 | 3.74 | 4.02 ± 0.03 |
| 3 | 13.26 | 30.0 | 7.48 | 3.57 | 3.51 ± 0.02 |

The significant results of each parameter and model verified by analysis of variance (ANOVA) are provided in Table 1. ANOVA showed that *A*, *B*, *C*, *AB*, *BC*, $A^2$, and $C^2$ of the model had significant effects on the biomass (Pr > *F* value less than 0.05). The failure to fit the term was not significant, indicating that the experimental design error was small and matched the actual situation. The *F* value of 36.57 and the $R^2$ value of 0.9524 reveal that the model fit is good and the confidence level is high, which could be used for analysis.

The Box–Behnken surface plot (Fig. S1) shows that the highest point predicted by the model is not located in the response center determined by the experiment, which could be due to the fact that the interaction among the factors is not considered in the single-factor optimization. Given the interaction of the factors having a greater impact on the experiment, it is necessary to carry out the validation of the model. The experimental prediction was performed based on the regression equation using Design Expert. Three representative groups were selected to experiment according to the conditions provided by the model, the results of which were presented in Table 2. The predicted experimental results are close to the actual experimental results, suggesting that the model has a high degree of confidence.

The optimal conditions obtained from the model equation were *A* (glucose) 14.88 g/L, *B* (tryptone) 37.65 g/L, and *C* (initial pH) 7.83. Under this culture, a final OD$_{600}$ of 4.02 was achieved, which is 81.1% higher than that of the mucin medium culture (Fig. 1A).

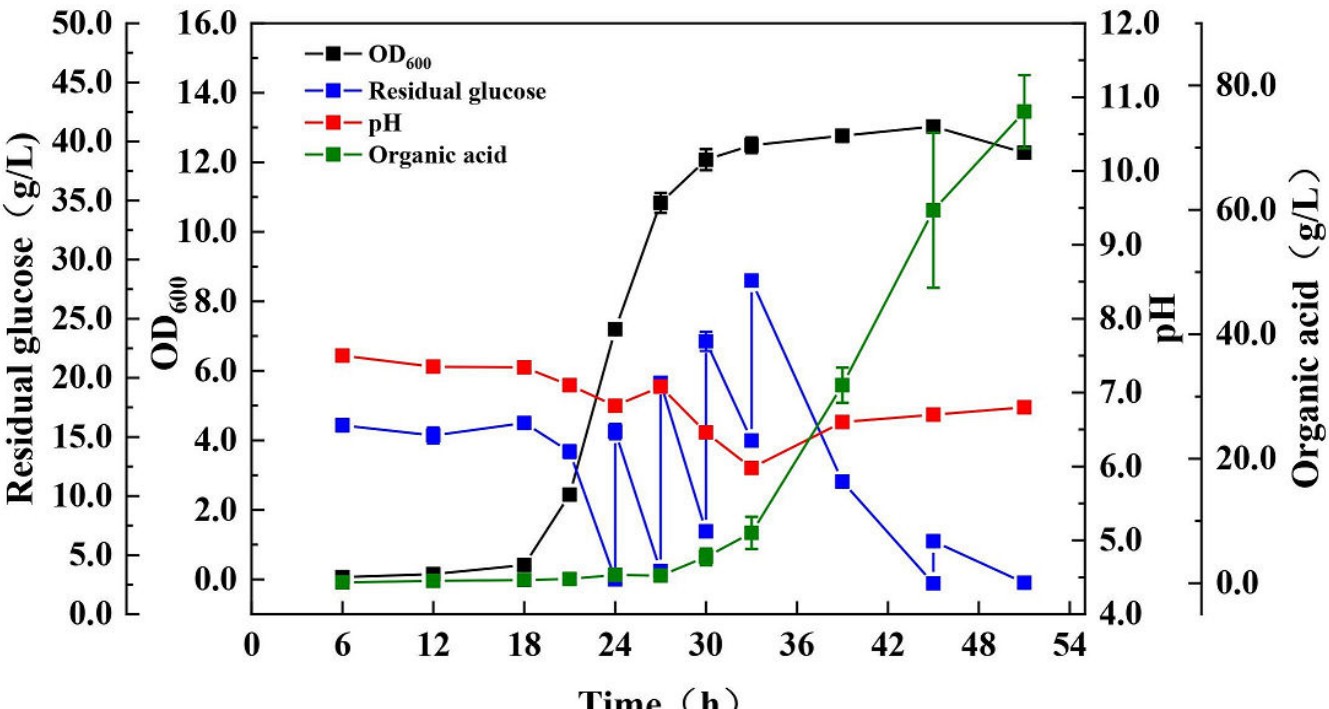

**FIG 3** The growth curve, pH change, organic acid production, and residual glucose consumption of *A. muciniphila* in a 5-L fermenter under the fed-batch strategy.

## Five-liter bioreactor for scale-up culture

In the 5-L bioreactor culture, the initial glucose of 15.0 g/L and the initial tryptone of 37.0 g/L were used as the carbon and nitrogen sources, respectively, and the concentrations of other components such as inorganic salts and vitamins were the same as the shake flask culture. Glucose was supplemented during the fermentation process according to the residual glucose concentration for the fed-batch culture.

As shown in Fig. 3, the exponential growth period of *A. muciniphila* ranged from 18 to 30 h, and the highest $OD_{600}$ of 13.03 was reached by 45 h, which is the highest reported value by using a sole carbon source (glucose). *A. muciniphila* performs anaerobic respiration, and its main metabolites are organic acids. The fermentation process consumed a total of 72.74 g/L glucose, while only 7.5 g/L of dry cell weight was reached. In the early stage, the cells grew in large quantities, the accumulation of organic acids was less, and the pH changed little. When a cell entered exponential growth, organic acids began to accumulate, and pH began to decrease. At this time point, glucose was fed in batches according the residual glucose concentration. During the whole exponential phase (18–30 h), the cells consumed 37.62 g/L glucose and accumulated 3.73 g/L organic acids, which indicates the supplemented glucose is mainly used for cell growth with less organic acid accumulation. The pH of this stage changes in the range of 7.8 and 7.0, suggesting that the initial pH of 7.8 is beneficial for cell growth. As the cells continue to grow, pH exhibited a fast decline from 26 to 33 h. After entering the stable period of 30–45 h, glucose consumed 25.62 g/L, while organic acids reached 51.85 g/L, indicating that the metabolites of organic acids gradually accumulate, i.e., the glucose consumed by *A. muciniphila* after 26 h is mainly used for the accumulation of organic acids. Considering that a certain amount of acetic acid and propionic acid has an anti-inflammatory ability (28, 29) that facilitates the function of *A. muciniphila* after gut colonization, the pH was adjusted to 6.5–7.0 after 33 h to reduce the inhibitory effect of organic acid accumulation, which maintained stable cell growth for a relatively long period without a sharp decline in the late fermentation period. These results demonstrate that this pH-regulating method could alleviate the growth inhibition of organic acids in *A. muciniphila,* and the fermentation strategy applied is quite effective for cultivating *A. muciniphila*.

According to our results, the biomass of *A. muciniphila* in the medium using glucose and tryptone as the carbon and nitrogen sources reached a higher value by expanding the culture in the 5-L bioreactor. Although tryptone is an animal-derived medium (hydrolyzed by casein from milk), it is cheaper than mucin and GlcNAc, which is convenient for scientific and clinical research. Future work will explore safer and low-cost media.

## Comparison of characteristics, protein expression, and supernatant metabolites of *A. muciniphila* under different media

### Effects of different media on bacterial morphology

Figure 4A illustrates that both cells have a similar morphology and size. They had an oval shape with a smooth cell surface and no flagellum or podocyte membrane. Their size was about (0.7–1.0) × (0.4–0.5) µm, which corresponds to the observation of Derrien et al. (13). These findings indicate that the optimized culture did not apparently change the morphology and the size of the cells.

### Effects of different media on cell hydrophobicity

*A. muciniphila* colonizes the mucus layer of the human intestine; the interaction between the mucin in the mucus layer and probiotics is one of the main reasons for its function (12). The mucus layer is a hydrophobic structure. The higher the cell hydrophobicity, the more favorable it is for *A. muciniphila* to accumulate in the mucus layer and colonize it (18) to perform its functions. As shown in Fig. 4B, the cell hydrophobicity of *A.*

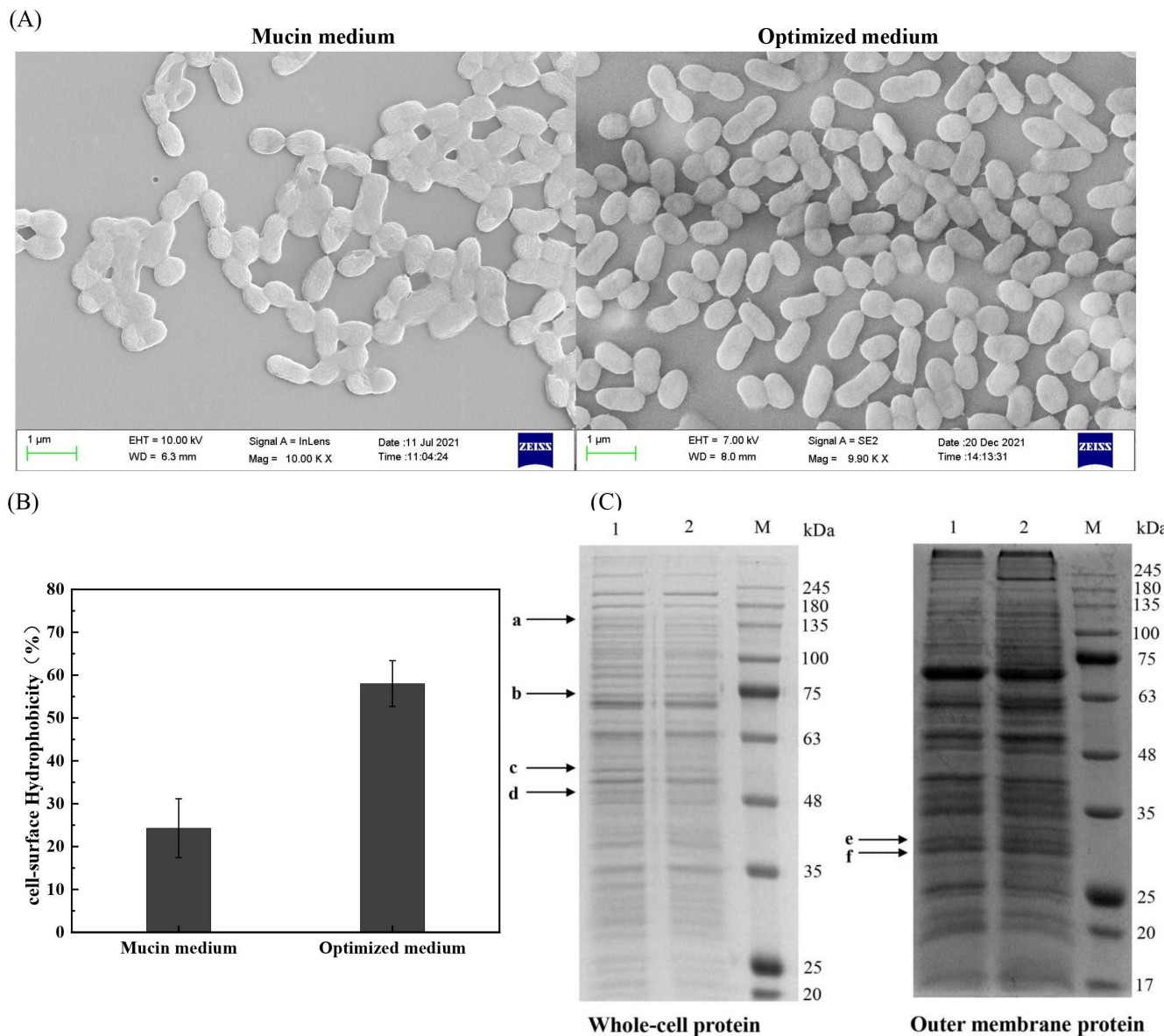

FIG 4  Comparison of cell morphology by scanning electron microscopy (SEM (A), protein expression (B) (lane 1: mucin medium, lane 2: optimized medium, and lane M: marker), and cell surface hydrophobicity (C) in different media during exponential growth stage.

*muciniphila* under the optimized medium was 2.38 times higher than that cultured in the mucin medium, indicating that *A. muciniphila* cultured in the optimized medium has

TABLE 3  Identification results of related protein bands

| Sample code | Protein | $-10\lg P$ | Coverage (%) | Spec | Avg. mass |
|---|---|---|---|---|---|
| a | Lactase | 436.12 | 56 | 580 | 142,051 |
| b | Sulfatase | 347.25 | 59 | 306 | 78,832 |
| c | Tryptophanase | 393.23 | 67 | 281 | 54,003 |
| d | Glycosyl hydrolase family 109 protein 1 | 316.29 | 72 | 242 | 53,803 |
| e | Amuc_1100 | 470.01 | 82 | 383 | 34,213 |
| f | Glucosamine-6-phosphate deaminase | 652.95 | 85 | 1,031 | 32,839 |

greater adhesion to the mucus layer, which may be more favorable for *A. muciniphila* to colonize the intestine and proliferate, whereby it interacts better with mucin and performs its function.

### Effects of different media on protein expression

The expressed whole-cell protein profile of *A. muciniphila* differed significantly in the two cultures, while the outer membrane protein profile was similar (Fig. 4C).

The four protein bands (a–d), which were differentially expressed in the whole-cell proteins, were cut and identified by LC-MS/MS. The most probable proteins were screened according to Spec parameters and are depicted in Table 3. Band c is identified as tryptophanase, whose lower abundance in the optimized medium indicates a reduced expression. Tryptophanase mainly catalyzes the anaerobic decomposition of tryptophan to produce indole, pyruvate, and ammonia and also has some effect on cysteine and serine. The mucin medium is rich in serine and cysteine, whereas the optimized medium did not contain them (30–32), leading to a decrease in tryptophanase expression. Mucin is a glycoprotein of high molecular weight with a complex structure, the breakdown of which is usually triggered by proteases acting on the non-glycosylated fraction, followed by glycosidases that cleave the α-glycosidic bonds at the terminal residues (33). Bands of a, b, and d are enzymes associated with mucin degradation. Their abundance became all lower in the optimized medium, suggesting that the expression of proteins associated with mucin degradation was reduced in the optimized medium. As the optimized medium included glucose and tryptone rather than mucin, it could not induce the expression of proteins associated with mucin degradation. Van Herreweghen et al. (34) cultured *A. muciniphila* in the human mimic gut without mucin for 8 days and then resupplied mucin to the culture, demonstrating that *A. muciniphila* could grow rapidly, which shows that *A. muciniphila* has strong adaptability (20) to mucin. When *A. muciniphila* recolonizes the human intestine, the expression of the associated mucin degradation proteins is trigged, promoting the growth of *A. muciniphila* using mucin.

*A. muciniphila* could express many outer membrane proteins involved in producing *A. muciniphila* pilus, regulating the host immune response, colonizing the host, and communicating with other microorganisms in the gut (35). It has been reported that the outer membrane protein Amuc_1100 (32 kDa) can interact with TLR2 to regulate both intestinal homeostasis and host metabolism (36–39). Therefore, due to its function, Amuc_1100 is considered one of the most important outer membrane proteins. The identification of two outer membrane protein bands (e and f) near 32 kDa in Fig. 4C is also shown in Table 3. It was found that band e contained Amuc_1100 protein, and the expression abundance in the two cultures was almost identical (Fig. 4C), indicating that *A. muciniphila* retains the relevant outer membrane functional protein under optimal culture conditions.

In conclusion, although the optimized culture reduced the expression of mucin degradation-related proteins, it did not appear to have any effect on the outer membrane proteins, showing that the optimized culture does not alter the main function of the cell.

### Effects of different media on supernatant small molecules

The metabolites of *A. muciniphila* are mainly organic acids, such as acetic acid and propionic acid, which confer beneficial functions to *A. muciniphila* in the intestinal tract. The other small molecules also play a crucial role in the metabolism of *A. muciniphila*.

Ninety-seven substances were identified by LC-MS, including organic acids, amino acids, sugars, and lipids (data not shown). OPLS-DA analysis performed using SIMCA software is illustrated in Fig. 5A. The metabolites of the samples could be clearly distinguished between the two groups. The biological reproducibility between the groups was good with small errors, indicating that the experiments were reliable and could be analyzed in the next step. The substances with VIP > 1 and *P*-value <0.5 were screened further, yielding a total of 21 small molecule metabolites with significant

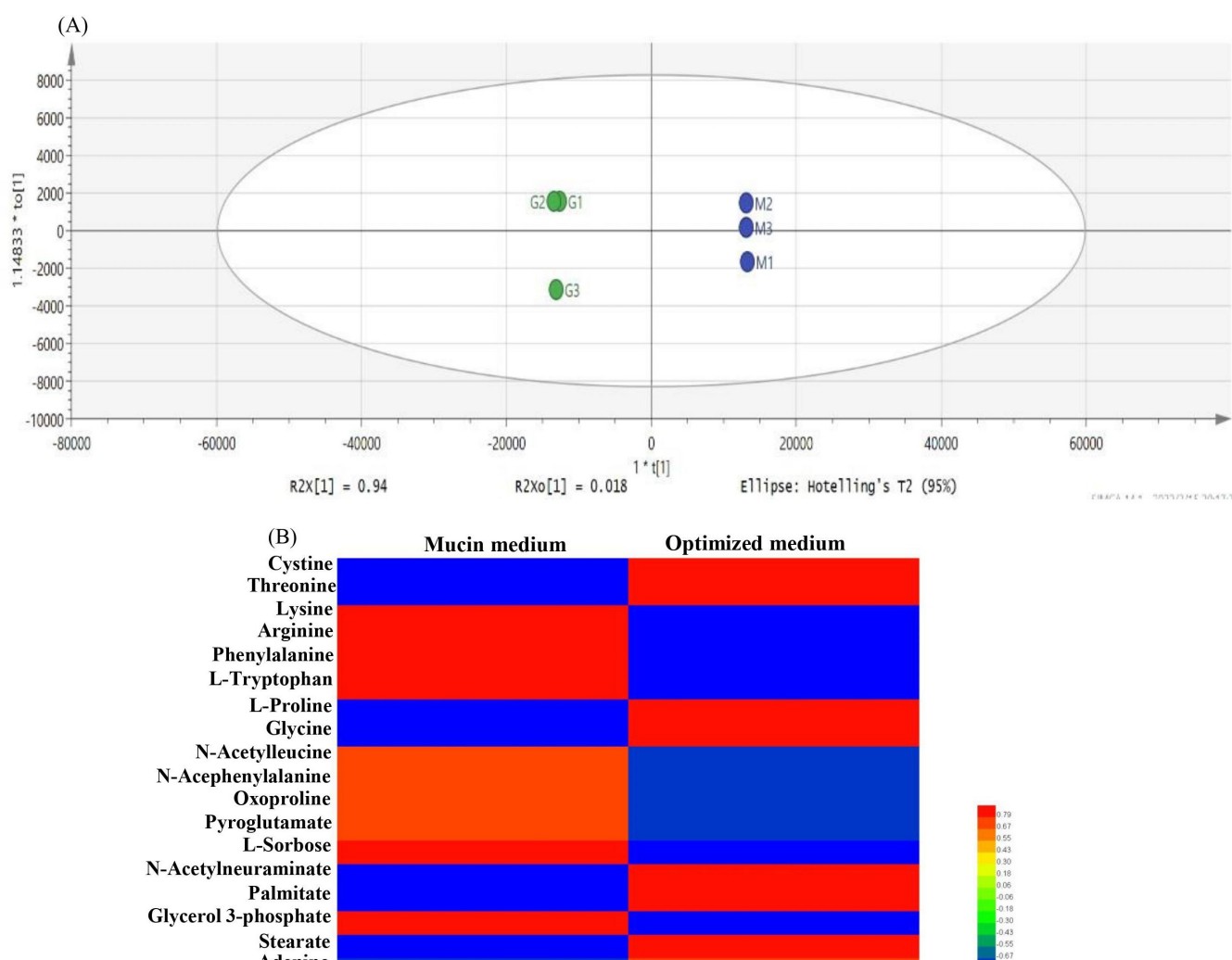

**FIG 5** Score scatter plot for OPLS-DA (A) (blue circle: mucin medium; green circle: optimized medium) and clustering of metabolites (B) (red and blue represent higher and lower levels of the metabolites, respectively).

differences (Table 4). Compared to the mucin medium, 13 substances were significantly up-regulated and eight substances were significantly down-regulated after optimized incubation.

Among the 21 substances, three substances of glutamate, oxoglutarate, and dethiobiotin were detected in the optimized medium but not in the mucin medium. Therefore, the remaining 18 significant substances that appeared in the two cultures were carried out in the cluster analysis (Fig. 5B), revealing that the substances with significant differences were mainly amino acids, lipids, and sugars.

The supernatant amino acids demonstrated a significant difference between the two cultures. The amino acids in the optimized medium are mainly provided by tryptones rich in glutamic acid, leucine, lysine, phenylalanine, and proline (30). In contrast, the amino acids in the mucin medium are mainly provided by mucins rich in threonine, serine, proline, and cysteine. Mucins are also carbon sources containing N-acetylneuraminic acid. Moreover, mucins contain hydrophobic regions that can bind lipids, such as palmitate and stearate (31, 32). Therefore, it is assumed that the differences in the amino acids, lipids, and some sugars in the supernatant are principally caused by the different compositions of the culture media.

**TABLE 4** Significantly different supernatant small molecules[a]

| Metabolites | Mucin medium | Optimized medium |
| --- | --- | --- |
| Cystine | 1,309.504 ± 28.036 | 0.678 ± 0.272 |
| Threonine | 462.237 ± 12.702 | 6.012 ± 0.338 |
| Lysine | 5.192 ± 4.147 | 167.465 ± 6.079 |
| Arginine | 2.047 ± 0.553 | 131.802 ± 5.169 |
| Phenylalanine | 107.020 ± 18.956 | 187.539 ± 7.232 |
| L-Tryptophan | 20.615 ± 4.004 | 86.108 ± 3.719 |
| L-Proline | 62.600 ± 2.550 | 3.935 ± 0.156 |
| Glycine | 51.967 ± 1.813 | 0.523 ± 0.005 |
| Glutamate | N/A | 1.602 ± 0.157 |
| N-Acetylleucine | 15.458 ± 2.668 | 96.443 ± 54.048 |
| N-Acetylphenylalanine | 7.020 ± 1.348 | 58.378 ± 30.553 |
| Oxoproline | 12.242 ± 2.630 | 71.697 ± 37.036 |
| Pyroglutamate | 12.242 ± 2.630 | 73.692 ± 33.583 |
| L-Sorbose | 0.063 ± 0.027 | 53.878 ± 2.055 |
| N-Acetylneuraminate | 111.662 ± 11.108 | 28.774 ± 1.915 |
| Glycerol 3-phosphate | 0.204 ± 0.100 | 50.527 ± 1.506 |
| Oxoglutaric acid | N/A | 0.024 ± 0.011 |
| Palmitate | 396.420 ± 29.567 | 1.081 ± 0.361 |
| Stearate | 156.453 ± 11.277 | 0.697 ± 0.032 |
| Adenine | 148.102 ± 75.839 | 0.182 ± 0.019 |
| Dethiobiotin | N/A | 0.714 ± 0.387 |

[a]The above values are expressed as peak area/(OD$_{600}$ × 10$^5$); data are shown as the mean ± SD of three biological replicates, and N/A means the substance was not detected.

As listed in Table 4, the supernatant metabolites from the optimized culture contain dethiobiotin, which was not detected in the mucin medium. Dethiobiotin is one of the vitamins that promote bacterial growth and reproduction. Some bacteria, including *Escherichia coli* and *Brevibacterium* sp., could use dethiobiotin since they can convert dethiobiotin to biotin by biotin synthase (BioB) (40). Biotin, also known as vitamin B7 or vitamin H, is a water-soluble multivitamin that is involved in fatty acid synthesis, sugar isomerization, and amino acid catabolism. In addition, it is involved in the regulation of gene expression, cell proliferation, repair of DNA damage, and stability of the chromatin structure through the biotinylation of histones, thus promoting bacterial growth (41–43). Dethiobiotin can be synthesized from heptanedioic acid catalyzed by 6-carboxyhexanoic acid-CoA ligase (BioW), 7-keto-8-aminononanoic acid synthase (BioF), and dethiobiotin synthase (BioD) (44, 45). Certain dethiobiotins are converted into biotin by BioB to promote cell proliferation, while the excess dethiobiotin is released to the medium. As shown in Fig. 6A, the content of heptanedioic acid reached the highest value at 36 h in the optimized medium, while no heptanedioic acid was detected in the mucin medium. Meanwhile, the relative expression levels of dethiobiotin metabolism-related genes in the optimized medium were significantly higher than that in the mucin medium (Fig. 6B), which also achieved the highest level at 36 h. It is conjectured that the optimized medium may induce cells to produce more heptanedioic acid, leading to the synthesis of dethiobiotin, which promotes cell growth and reproduction. It is conjectured that the optimized medium may induce cells to produce more heptanedioic acid, leading to the synthesis of dethiobiotin, which promotes cell growth and reproduction.

Moreover, the content of glycerol 3-phosphate in the optimized medium was much higher than that in the mucin medium. The glycolytic pathway of *A. muciniphila* in the optimized medium uses glucose to produce glyceraldehyde 3-phosphate and dihydroxyacetone phosphate. Glyceraldehyde 3-phosphate continues to participate in the glycolytic pathway by the action of GAPDH, while dihydroxyacetone phosphate can be reversibly converted to glyceraldehyde 3-phosphate by triose phosphate isomerase (Amuc_0562) or converted to glycerol 3-phosphate by GPDH. Figure 6C presents that the

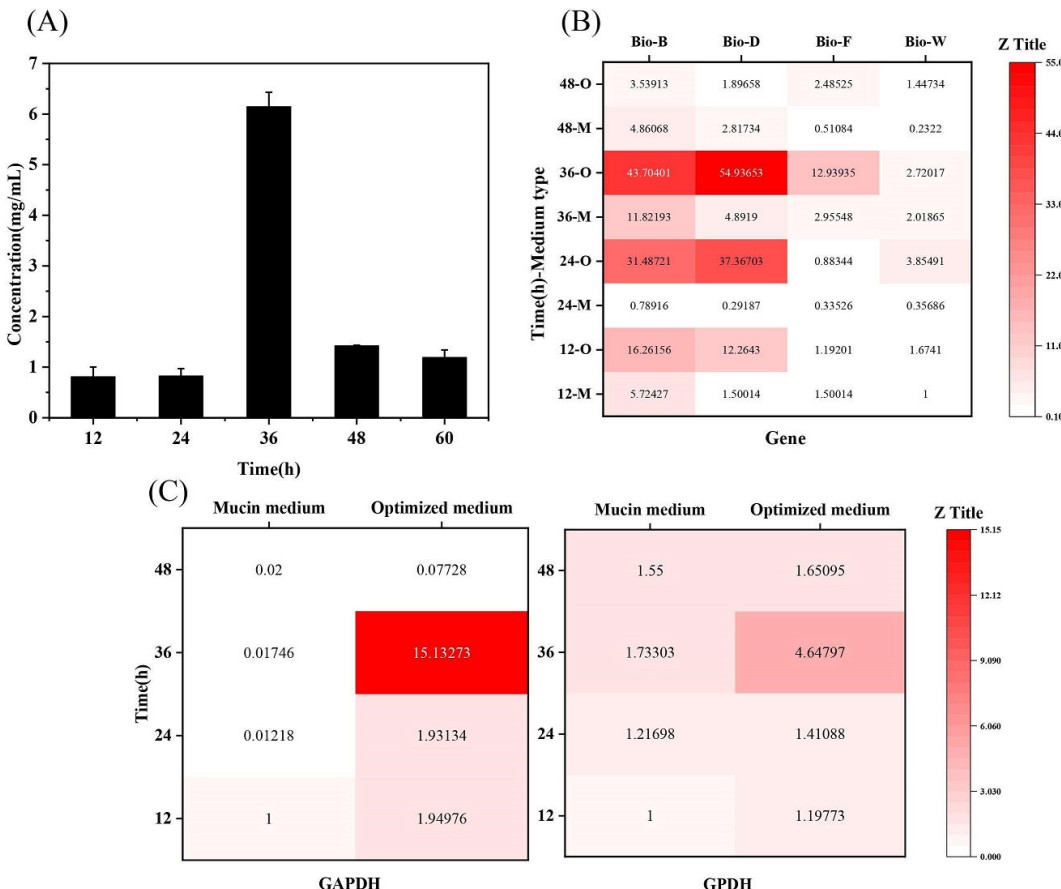

**FIG 6** Changes in the concentration of the metabolite heptanedioic acid in the optimized medium at different time points (A) and the relative expression levels of pimelic acid metabolism genes (B) and gene expression levels of glyceraldehyde 3-phosphate dehydrogenase (GAPDH)/glycerol 3-phosphate dehydrogenase (GPDH) (C) in the optimized medium and mucin medium at different time points (M: mucin medium; O: optimized medium, the point whose data is "1" in the figure is the standard).

transcriptional expression level of GAPDH and GPDH genes in the optimized medium was much higher than that in the mucin medium, especially at 36 h. GAPDH is not only present in the cytoplasm, cell membrane, and mitochondria but is also distributed on the cell surface as an adhesion molecule to play a partial role in adhesion, which may interact with other bacterial surface components by charge or hydrophobicity (46). As illustrated in Fig. 4B, cell hydrophobicity was clearly enhanced in the optimized medium, which may be because more intracellular GAPDH being secreted at the cell surface improves cell adhesion by hydrophobicity. The secretion of GAPDH to the cell surface resulted in the intracellular accumulation of glyceraldehyde 3-phosphate that is converted to dihydroxyacetone phosphate, which is further converted to glycerol 3-phosphate by GPDH. As a result, it is presumed that the optimized culture may increase the secretion of GAPDH at the cell surface through the metabolic regulation of the glycerol 3-phosphate pathway, thereby improving cell surface hydrophobicity and adhesion to mucin.

## Conclusions

*A. muciniphila* has a short discovery history and strict requirements for *in vitro* culture. This study achieved a maximum $OD_{600}$ of 13.03 ($1.03 \times 10^{10}$ CFU/mL) in the bioreactor under optimum conditions, which has been the highest level reported by the use of a sole carbon source (glucose). Moreover, the optimized culture had no significant effect on bacterial morphology and the outer membrane functional proteins, while it could obviously improve cell hydrophobicity, which is more conducive to cell colonization of

the gut. In future work, the production of *A. muciniphila* is expected to increase by nitrogen feeding and medium replacement to meet the needs for economical production.

## MATERIALS AND METHODS

### Strain, medium, and culture conditions

*A. muciniphila* (ATTC BAA-835) obtained from the American Type Culture Collection (USA) was resuscitated in BHI medium to logarithmic phase and inoculated into the basal medium with 4% innoculation to be cultured anaerobically, in which cysteine replaced $Na_2S$, as previously described (13). The ingredients of this basal medium contains (L) $NaHCO_3$ 4.0 g, $KH_2PO_4$ 0.4 g, $Na_2HPO_4 \cdot 12H_2O$ 1.34 g, $NH_4Cl$ 0.3 g, NaCl 0.3 g, $MgCl_2 \cdot 6H_2O$ 0.1 g, $CaCl_2$ 0.11 g, L-cysteine 0.4 g, vitamin solution 1 mL, acid trace solution 1 mL, and alkali trace solution 1 mL, medium pH adjusted to 7.0. $NaHCO_3$, threonine, and vitamins were filtered by a 0.22-µm sterile filter membrane to remove bacteria. The mucin medium, referred to as 8% mucin (vol/vol), is added to the basal medium. Unless indicated, incubations were performed in serum bottles sealed with butyl rubber stoppers at 37°C under anaerobic conditions provided by a gas phase of $N_2/CO_2$ (80:20, vol/vol).

### Optimization of carbon and nitrogen sources, buffer systems, and the original pH

The optimum carbon was screened out in the basal medium, in which tryptone was used as the nitrogen source, and the carbon sources of glucose, GlcNAc, oligogalactose, fucose, mannose, xylose, galactose, sucrose, and glycerol were investigated, respectively. The optimal nitrogen was also determined in the basal medium, in which glucose was used as the carbon source, and the nitrogen sources of tryptone, YE, BE, peptone, PP, and SP were investigated, respectively. Subsequently, the selection of the buffer systems of MOPS, $KH_2PO_4/Na_2HPO_4 \cdot 12H_2O$, and $NaHCO_3$ was performed. The optimum original pH was determined between 6.5 and 8.5, with intervals of 0.5 pH units (adjusted with HCl or NaOH).

### Scale-up culture in the 5-L bioreactor

The 5-L bioreactor culture was carried out by fed-batch and stage pH controlling, and the amount of inoculation in fermenter is 10%. The anaerobic gas ($N_2:CO_2$, 8:2) was continuously ventilated during the culture process. The original pH of the medium was controlled at 7.8 using a solution of NaOH. As *A. muciniphila* grew, the pH in the medium gradually decreased. When the pH of the medium dropped to 6.0, it was regulated in the range of 6.5–7.0. Samples were taken at certain intervals, and glucose was replenished according to glucose consumption.

### Biomass determination

#### *Cell density test of $OD_{600}$*

The absorbance values were measured at 600 nm using a spectrophotometer after the bacterial solution had been diluted several times. As a blank control, the corresponding liquid medium without inoculation was used.

### Determination of viable bacteria

The cells were collected by centrifugation at 6,000 × *g* for 5 min at 4°C and then washed twice with sterile PBS. After a gradient dilution with sterile PBS solution, 50 µL of the diluted bacterial solution was evenly applied to a solid plate, placed in an anaerobic culture box, and incubated at 37°C for 48 h. The colony number (*A*) was taken out and calculated as follows:

$$\text{Viable bacteria (CFU/mL)} = A \times \text{Dilution times} \times 20$$

To convert between $OD_{600}$ and CFU, the following formula was used:

$$\text{CFU}(10^8/\text{mL}) = 8 \times OD_{600} - 0.9$$

## Determination of glucose concentration

The supernatant obtained by centrifugation at 10,000 $\times$ $g$ for 3 min was diluted at a certain gradient and then submitted to the SBA Biosensor Analyzer (SBA-40E, Shandong) for the measurement of the residual glucose concentration.

## Observation of cell morphology

Cell morphology was observed by SEM analysis according to the reported method by Li et al. (6).

## Extraction of whole-cell lysate

Ten milliliters of the late stage of exponential growth ($OD_{600}$ about 3.0) bacteria were collected, centrifuged at 10,000 $\times$ $g$ at 4°C for 3 min, washed with prechilled PBS buffer twice, resuspended in prechilled 0.2 mol/L PBS solution or 8 M urea solution, and sonicated under ice bath, with 15% crushing power, a time interval of 3:6 (s/s), and a crushing time of 15 min. The lysate was centrifuged at 10,000 $\times$ $g$ for 30 min at 4°C to extract the supernatant.

## Outer membrane protein extraction and concentration determination

The cell pellets of *A. muciniphila* harvested after 48 h were subjected to outer membrane protein extraction using the bacterial outer membrane protein extraction kit (Bestbio BB-31512, Shanghai, China). The protein concentration was estimated with the BCA Protein Assay Kit (Sangon Biotech C503021, Shanghai, China).

## SDS-PAGE and LC-MS/MS analysis

After pretreatment, both intracellular and membrane protein samples were added to 4× SDS-PAGE loading buffer with DTT (Solarbio Life Science, Beijing, China) and boiled for 10 min. The proteins were separated on 10% SDS-PAGE and visualized using Coomassie Brilliant Blue (20). ColorMixed Protein Marker 5–245 kD (Solarbio Life Science, Beijing, China) was used to determine the apparent molecular weight of the separated proteins. The differentially expressed proteins were cut and identified by LC-MS/MS (Bruker timsTOF Pro, Germany).

## Hydrophobicity analysis

One milliliter of cell suspension was washed with PBS buffer twice, and the cell pellets were resuspended in 1 mL of PBS buffer, the $OD_{600}$ of which was measured ($OD_{initial}$). Subsequently, it was mixed with an equal volume of xylene and vortexed for 5 min. The $OD_{600}$ of the aqueous phase ($OD_{aqueous\ phase}$) was measured after the solution was fully stratified. The hydrophobicity of *A. muciniphila* was calculated as follows:

$$\text{Hydrophobicity} = (1 - OD_{initial} \div OD_{aqueous\ phase}) \times 100\%$$

## Determination of organic acids of metabolites

The fermentation broth (1 mL) was collected and slowly mixed with 1 M hydrochloric acid (1:1, vol/vol), vortexed, and passed through a 0.22-µm filter. The supernatant was injected into the Agilent 7890 gas chromatograph equipped with an electron capture detector to determine the organic acids of *A. muciniphila*.

The following gas chromatography conditions were applied: instrument, Agilent-7890A; chromatography column, DB-FFAP 122-3232; detector temperature, 250℃; inlet temperature, 250℃; split ratio, 10:1; heating program, between 80℃ and 200℃ for 25 min; and injection volume, 1 µL.

## Measurement of supernatant small molecules

Four hundred microliters of MeOH and 400 µL of ACN were added to a 100-µL supernatant collected from the fermentation broth, placed in a vortex for 30 s, and then sonicated for 10 min under an ice bath with 15% crushing power and time interval of 3:6 (s/s). The lysate was centrifuged for 15 min at $13,000 \times g$ and 4℃. The supernatant was snap-frozen with liquid nitrogen after storage at −80° and dried in a vacuum freeze dryer for 24 h. The freeze-dried powder was sent to the Core Facility of Biomedical Sciences, Xiamen University, for testing using LC-MS (AB Sciex TripleTOF 5600+, Singapore).

## Quantitative real-time PCR

Total RNA was extracted using the MiniBEST RNA extraction kit (Takara Bio Inc., Beijing, China), and then, 200 ng/µL of RNA was reverse-transcribed to the complementary DNA (cDNA) using the Easy-Script First-Strand cDNA Synthesis SuperMix kit (TransGen Biotech, Beijing, China). The quantitative real-time PCR reaction was carried out using Trans-Start Top Green qPCR SuperMix (TransGen Biotech, Beijing, China). The reaction without cDNA template was used as a negative control, and the 16s rRNA gene was used as a reference gene. The primers used are listed in Table S3. The relative expression levels of each gene were calculated using the $2^{-\Delta\Delta CT}$ method.

## Statistical analysis

Data were expressed as the mean ± SD (the standard deviation). The data in this experiment represent the average values of three independent repeated experiments, and the error bars represent standard errors. Origin 2018 software was used to draw the graphs. The response surface experiments were designed using Design Expert.

## ACKNOWLEDGMENTS

We gratefully acknowledge the Fujian Collaborative Innovation Center for Exploitation and Utilization of Marine Biological Resources for its continuous support. We would like to thank MogoEdit for its English editing during the preparation of this manuscript.

This work was supported by the Natural Science Foundation of Xiamen, China (no. 3502Z20227183), the National Key Research and Development Program of China (no. 2022YFC2104600), and the National Natural Science Foundation of China (no. 31871779).

## AUTHOR AFFILIATIONS

[1]Department of Chemical and Biochemical Engineering, College of Chemistry and Chemical Engineering, Xiamen University, Xiamen, People's Republic of China
[2]Xiamen Key Laboratory of Synthetic Biotechnology, Xiamen University, Xiamen, People's Republic of China
[3]The Key Laboratory for Chemical Biology of Fujian Province, Xiamen University, Xiamen, People's Republic of China
[4]Department of Microbiome and Health, Bluepha Co., Ltd, Shenzhen, People's Republic of China

## AUTHOR ORCIDs

Xueping Ling 🔟 http://orcid.org/0000-0002-6357-8681

## FUNDING

| Funder | Grant(s) | Author(s) |
|---|---|---|
| 厦门市科学技术局 \| Natural Science Foundation of Xiamen Municipality (Xiamen Natural Science Foundation) | 3502Z20227183 | Xueping Ling |
| MOST \| National Key Research and Development Program of China (NKPs) | 31871779 | Xueping Ling |
| MOST \| National Natural Science Foundation of China (NSFC) | 31871779 | Ning He |

## AUTHOR CONTRIBUTIONS

Haiting Wu, Software, Validation, Visualization, Writing – original draft, Writing – review and editing | Shuhua Qi, Data curation, Methodology, Software, Visualization, Writing – original draft | Ruixiong Yang, Validation, Visualization, Writing – original draft | Qihua Pan, Validation | Yinghua Lu, Data curation, Funding acquisition, Investigation, Resources | Chuanyi Yao, Conceptualization, Data curation, Investigation | Ning He, Funding acquisition, Investigation | Song Huang, Investigation, Methodology | Xueping Ling, Conceptualization, Funding acquisition, Project administration, Resources, Supervision, Writing – original draft, Writing – review and editing

## DATA AVAILABILITY

The authors will supply the relevant data in response to reasonable requests.

## ADDITIONAL FILES

The following material is available online.

### Supplemental Material

**Supplemental material (Spectrum02386-23-s0001.docx).** Tables S1 to S3 and Fig. S1.

### Open Peer Review

**PEER REVIEW HISTORY (review-history.pdf).** An accounting of the reviewer comments and feedback.

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
