## [Reviewer comments · Microbiology Spectrum]

Microbiology Spectrum

Strategies for high cell-density cultivation of *Akkermansia muciniphila* and its potential metabolism

Haiting Wu, Shuhua Qi, Ruixiong Yang, Qihua Pan, Yinghua Lu, Chuanyi Yao, Ning He, Song Huang, and Xueping Ling

Corresponding Author(s): Xueping Ling, Xiamen University College of Chemistry and Chemical Engineering

Review Timeline:

Submission Date:	June 20, 2023
Editorial Decision:	September 20, 2023
Revision Received:	October 12, 2023
Accepted:	November 8, 2023

Editor: Erik Hom

Reviewer(s): Disclosure of reviewer identity is with reference to reviewer comments included in decision letter(s). The following individuals involved in review of your submission have agreed to reveal their identity: Lu-Jing Ren (Reviewer #1)

Transaction Report:

DOI: <https://doi.org/10.1128/spectrum.02386-23>

September 20, 2023

Dr. xueping Ling
Xiamen University College of Chemistry and Chemical Engineering
422 Siming South Road
Xiamen
China

Re: Spectrum02386-23 (Strategies for high cell-density cultivation of *Akkermansia muciniphila* and its potential)

Dear Dr. xueping Ling:

Thank you for submitting your manuscript to Microbiology Spectrum. Given the prior reviews of your manuscript at AEM, I secured an additional reviewer who suggests several additional minor revisions. When submitting the revised version of your paper, please provide (1) point-by-point responses to the issues raised by the reviewers as file type "Response to Reviewers," not in your cover letter, and (2) a PDF file that indicates the changes from the original submission (by highlighting or underlining the changes) as file type "Marked Up Manuscript - For Review Only". Please use this link to submit your revised manuscript - we strongly recommend that you submit your paper within the next 60 days or reach out to me. Detailed instructions on submitting your revised paper are below.

Link Not Available

Sincerely,

Erik Hom

Journals Department
Reviewer comments:

Reviewer #1 (Comments for the Author):

The article by Wu et. al. systematically developed an economical high cell-density cultivation method of *A. muciniphila*, and proposed that the optimized culture had no negative effect on their colonization ability. This work could be very useful for the future commercialization of *A. muciniphila*.

- 1.The panel number in Fig. 2 is not consistent with the description of the main text. Please correct accordingly;
- 2.In Page 2 , Line 24: "*Akkermansia*" should be in italics.
- 3.It is very interesting that the pH increased significantly during the lag phase if the initial pH was lower than 8.5 (Fig. 2).
 - a)What is the concentration of the buffer agent used?

b)How do you prepare the inoculum and what is the inoculum size?

c)Since the cell density is very low during the lag phase, the bacteria must produce tons of alkaline metabolites to achieve such a significant change from initial pH especially with acidic initial pH. Do you have any explanation why this would happen?

d)It looks like that in the 5-L scale-up batch, the pH did not increase during the first 12 h. Do you have any explanation why this is different compared with flask culture?

4.The name of heptanedioic acid used is not consistent in the main text and Fig. 6 caption. The author should use a same name when refer to the same metabolite to avoid any confusion.

5.The author found that when cultured in optimized media, GAPDH expression level increase and cell hydrophobicity also increased. Though surface GAPDH may increase the adhesion ability, there is no evidence showed that GAPDH is related with cell hydrophobicity. Therefore, calling GAPDH and GPDH cell hydrophobicity genes in Fig.6 caption is inappropriate.

Overall, I would recommend to publish if the above minor concerns are addressed.

Staff Comments:

Preparing Revision Guidelines

Please return the manuscript within 60 days; if you cannot complete the modification within this time period, please contact me. If you do not wish to modify the manuscript and prefer to submit it to another journal, please notify me of your decision immediately so that the manuscript may be formally withdrawn from consideration by Microbiology Spectrum.

Response to Reviewer's Comments

Dear reviewers,

We would like to thank you for your kind review and comments on the manuscript Spectrum02386-23, which greatly helped to improve the quality of our work. The paper has been carefully revised accordingly (as shown by "red color" in the revision), and the point-to-point reply is arranged below in blue as follows.

Comments from reviewer 1:

1. The panel number in Fig. 2 is not consistent with the description of the main text. Please correct accordingly;

Response: Thanks for your careful correction, and we are so sorry for our mistake. The error has been modified in Fig.2 of the revised manuscript (Page 38).

2. In Page 2, Line 24: "Akkermansia" should be in italics.

Response: Thanks for your careful correction, and we are so sorry for our mistake. The error has been modified in the revised manuscript (Page 2, line 2).

3. It is very interesting that the pH increased significantly during the lag phase if the initial pH was lower than 8.5 (Fig. 2).

a) What is the concentration of the buffer agent used?

b) How do you prepare the inoculum and what is the inoculum size?

c) Since the cell density is very low during the lag phase, the bacteria must produce tons of alkaline metabolites to achieve such a significant change from initial pH especially with acidic initial pH. Do you have any explanation why this would happen?

d) It looks like that in the 5-L scale-up batch, the pH did not increase during the first 12 h. Do

you have any explanation why this is different compared with flask culture?

Response: Thank you for your valuable comments.

- a) The buffer solutions used in this experiment are as follows: KH_2PO_4 0.4 g/L, $\text{Na}_2\text{HPO}_4 \cdot 12\text{H}_2\text{O}$ 1.34 g/L, and NaHCO_3 4.0 g/L (Page 22, line 11).
- b) *A. muciniphila* resuscitated in BHI medium to logarithmic phase were inoculated into fermentation medium. The amount of inoculation in shake flask is 4%, and that in fermenter is 10%.
- c) Previous studies (24, 25) and the present work (Fig. 2A) have shown that a slightly alkaline environment is more favorable for *A. muciniphila* growth. In order to adapt to the growing environment, alkaline substances may be metabolized by cells in lag phase to make it slightly alkaline for optimal growth of the organism. Specific alkaline substances will be further analyzed in the subsequent research.
- d) As shown in Fig.3, the original pH of the medium in fermenter was controlled at 7.8, which was the optimum pH for cell growth. Fig. 2B showed that when the initial pH in shake flask was lower than 8.5, the lower the initial pH, the greater the increase in the medium pH, and the final lag phase pH was concentrated in the range of 7.3 to 8.2. Therefore, the initial pH of 7.8 did not make obvious change in the medium pH during the lag phase.

4. *The name of heptanedioic acid used is not consistent in the main text and Fig. 6 caption. The author should use a same name when refer to the same metabolite to avoid any confusion.*

Response: We appreciate your invaluable and constructive comments. The consistent name of “heptanedioic acid” was used through the main text and captions.

5. *The author found that when cultured in optimized media, GAPDH expression level increase and cell hydrophobicity also increased. Though surface GAPDH may increase the adhesion ability, there is no evidence showed that GAPDH is related with cell hydrophobicity. Therefore, calling GAPDH and GPDH cell hydrophobicity genes in Fig.6 caption is inappropriate.*

Response: Thank you for seeking this valuable clarification to enhance clarity. The description of cell hydrophobicity genes has been modified to “gene expression levels of GAPDH/GPDH” in caption (Page 32, line 19).

The authors highly appreciated all the comments from the reviewers which have made this manuscript much improved.

Re: Spectrum02386-23R1 (Strategies for high cell-density cultivation of Akkermansia muciniphila and its potential metabolism)

Dear Dr. Xueping Ling:

I will be accepting your manuscript, and I am forwarding it to the ASM production staff for publication. Your paper will first be checked to make sure all elements meet the technical requirements. ASM staff will contact you if anything needs to be revised before copyediting and production can begin. Otherwise, you will be notified when your proofs are ready to be viewed.

IMPORTANT: in the proofs stage, please make sure you add modify the text to incorporate your response to the reviewer's comments, specifically, your response:

"b) A. muciniphila resuscitated in BHI medium to logarithmic phase were inoculated into fermentation medium. The amount of inoculation in shake flask is 4%, and that in fermenter is 10%."

This is important information that should have been included in your revised manuscript. However, as this is minor, I will not hold up acceptance and ask that you please make this modification.

Sincerely,
Erik Hom
Editor
Microbiology Spectrum